# Integrated malaria service delivery and its determinants among pregnant women in Ethiopia: Multi level analysis of 2021/22 National Service Provision Assessment Survey

Kassawmar Angaw Bogale[1]*, Kassahun Alemu[2], Kindie Fentahun Muchie[1,3], Mulusew Andualem Asemahagn[4], Hailemariam Awoke Engedaw[5], Muluken Azage Yenesew[6]

1 Department of Epidemiology and Biostatistics, School of Public Health, College of Medicine and Health Sciences, Bahir Dar University, Bahir Dar, Ethiopia, 2 Department of Epidemiology and Biostatistics, Institute of Public Health, College of Medicine and Health Sciences, University of Gondar, Gondar, Ethiopia, 3 Department for Infectious Disease and Tropical Medicine, University Hospital Heidelberg, Heidelberg, Germany, 4 Department of internal medicine, School of Medicine, College of Medicine and Health Sciences, Bahir Dar University, Bahir Dar, Ethiopia, 5 Department of Environmental Health, School of Public Health, College of Medicine and Health Sciences, Bahir Dar University, Bahir Dar, Ethiopia, 6 Department of Health system and health Economics, School of Public Health, College of Medicine and Health Sciences, Bahir Dar University, Bahir Dar, Ethiopia

* kassawmarbogale@gmail.com

## Abstract

### Background

Malaria in pregnancy poses significant health risks to mothers and their, fetuses, and newborns children in Tropical and subtropical countries including Ethiopia. The delivery of integrating essential malaria services into routine antenatal care is crucial for effective prevention and control. However, evidence on the extent and determinants of this integration in Ethiopia remains limited.

### Objective

This study aimed to assess the delivery of integrated malaria services during ANC visit and its determinants among pregnant women in Ethiopia.

### Methods

We conducted a secondary analysis of the Ethiopia Service Provision Assessment Survey 2021/22, a nationally representative cross-sectional study. The final sample included 4273 pregnant women nested across 662 health facilities. Factors were identified based on the WHO Malaria in Pregnancy framework. Multilevel logistic regression models were applied to identify significant factors influencing integrated service uptake.

**Data availability statement:** The data used in this study are publicly available from the Demographic and Health Surveys (DHS) Program. Specifically, the analysis was based on the Ethiopia Service Provision Assessment (SPA) Survey 2021/22. The dataset is anonymized and accessible upon registration and approval through the DHS Program data portal: https://dhsprogram.com/data/.

**Funding:** The author(s) received no specific funding for this work.

**Competing interests:** The authors have declared that no competing interests exist.

## Result

Only 7.9% of pregnant women attended ANC visits where all components of integrated malaria services were delivered concurrently, with substantial regional disparities. At the client level, women with two previous pregnancies (AOR = 1.67, 95% CI: 1.06–2.62), attending three or more ANC visits (AOR = 1.58, 95% CI: 1.04–2.40) and client who received an Insecticide-Treated Net during ANC (AOR = 2.81, 95% CI: 1.29–6.12) were more likely to attend ANC visits in which integrated malaria services were delivered. Furthermore, clients attending facilities with malaria-trained providers were more likely to receive integrated malaria services during ANC than those attending facilities without such training (AOR = 4.24, 95% CI: 1.80–10.00). Rural facility attendance was also positively associated with integrated malaria service delivery compared with urban facility attendance (AOR = 2.73, 95% CI: 1.04–7.19).

## Conclusion

Integrated malaria service delivery during ANC remains unacceptably low in Ethiopia, constrained by regional disparities and multilevel factors. Strengthen continuity of ANC follow up, updating policy on ITN distribution, strengthening providers' capacity, and addressing geographic disparities to accelerate progress toward WHO maternal health targets.

---

## Background

Malaria infection during pregnancy (MiP) continues to be a significant global public health concern, particularly in sub-Saharan Africa, Southeast Asia, and parts of Latin America. Its prevalence in sub-Saharan Africa is as high as 60%, with placental malaria affecting up to 28% of cases [1,2].

MiP poses a serious threat to the health of mothers, fetuses, and newborns ranging from asymptomatic cases to severe anemia and maternal death. Physiological changes during pregnancy increase vulnerability to malaria, leading to unique challenges and severe health risks [3,4]. Clinical manifestations in pregnant women include joint pain, fever, headaches, fatigue, nausea, and anemia, with complications varying across trimesters [3,5]. Beyond maternal health, MiP contributes to unfavorable birth outcomes, including miscarriage, premature delivery, and neonatal death. It can impair placental development and function, leading to poor fetal growth, intrauterine growth restriction, and long-term infant health problems. Even in areas with low-to-moderate transmission, MiP contributes significantly to adverse outcomes including stillbirth, neonatal death, and placental insufficiency [4,6,7].

The burden of MiP is further exacerbated by socioeconomic barriers, delayed care-seeking, and limited access to integrated healthcare services delivery. Inadequate diagnosis and mismanagement of malaria cases, along with provider limitations in counseling and clinical skills, compound the challenges in effectively

delivering MiP services [8]. For instance in Sub-Saharan Africa where the high malaria burden and inadequate malaria service during pregnancy reported that Low birth weight (LBW) related to malaria infection during pregnancy was accounted 35% of live births and 11% of neonatal mortality [2].

Ethiopia, home to over 132 million people, remains malaria-endemic, with 75% of its landmass conducive to transmission and an estimated 70% of pregnant women at risk [9]. The country aims to eliminate malaria among this group by 2030. While national malaria control efforts achieved substantial progress between 2000 and 2019, recent data indicate that malaria became a major public health problem for pregnant women [5,10–12].

Integration of malaria services delivery within antenatal care (ANC) is an essential strategy for improving prevention, early diagnosis, and management of malaria in pregnancy. Effective integration requires not only client engagement but also the simultaneous availability of preventive counseling, trained providers, and facility capacity to diagnose and treat malaria. According to the WHO framework, malaria service integration delivery with ANC is defined as the receipt of ANC in a setting where malaria prevention counseling, provider capability, and facility readiness were simultaneously available during the visit. [13–17].

Despite the several established advantages of malaria service integration with ANC [14–16,18–23], multilevel factors affect the integration of malaria service delivery into ANC services which leads low uptake of the service by pregnant women. From the system level factors; some countries including Ethiopia did not adopt the inclusion of IPTP and provision of ITN to their health policies [24] which affects the implementation of service integration. The availability, readiness, and actual provision of services at the facility level also affect the delivery of integrated MiP services [21,22,25–28]. The shortage of essential commodities and limited capacity of health professional to provide malarias service also affect the implementation of service integration [22,29,30]

Beyond structural and facility level influences, the effective delivery of integrated service is also shaped by client-level factors such as age, ANC follow-up adherence, pregnancy status, educational status, and awareness of malaria infection and prevention strategies [10,29,31–33].

While the WHO strongly recommends the delivery of integrating malaria services into ANC to improve prevention and treatment services, and despite malaria remaining endemic in several Ethiopian regions, evidence on the extent and the specific determinants influencing the delivery of integrated malaria service with ANC service remains limited. This study, therefore, uses nationally representative data from the 2021/2022 SPA survey to assess the level of integrated malaria services delivery with ANC and to identify the client and facility-level factors associated with such delivery of integration.

## Methods

### Data source and setting

Data were derived from the SPA Survey conducted in 2021/22 in Ethiopia. The SPA survey is a nationally representative, facility-based survey conducted by the Ethiopian Public Health Institute (EPHI) and Inner City Fund (ICF) [34]. The survey employed stratified multistage sampling to collect data from health facilities, health workers, and clients, designed to provide information on the availability and quality of health services at various levels of the healthcare system. All clients (pregnant women) visit the sampled health facilities during data collection were interviewed. The study was conducted across Ethiopia's nine regions and two city administrations.

Data were collected through structured questionnaires administered to health facility representatives, service providers, and exit interviews with pregnant women.

### Study population and sampling

The study population included pregnant women receiving ANC care services, along with their healthcare providers and the facilities where these services were rendered. The SPA Survey employed a multi-stage cluster sampling design to select

a nationally representative sample of health facilities [34]. The survey selected a stratified random sample of 1,407 health facilities, selected with probability systematic sampling. However, data was successfully collected from 1156 facilities; the remaining were permanently closed, not yet operational, under security issues, unreachable, or duplicates of another facility in the sample.

From a total of 1156 initial facilities, 662 facilities were ultimately nested in the analysis. The other 494 health facilities were excluded because of they did not provide ANC services, and thus no data were collected on ANC or malaria service integration from these facilities. The number of eligible clients identified and present for ANC services was 4355. From this initial client pool, 4273 clients were nested within 662 facilities. The final analysis, therefore, was based on 4273 clients, nested within in 662 facilities (Fig 1).

## Variables and measurements

Integrated malaria service delivery during antenatal care (ANC) was assessed using data from the 2021–22 Ethiopia Service Provision Assessment (SPA) survey. The SPA employs standardized instruments developed by the DHS Program and collects information at three complementary levels: client exit interviews, health care provider interviews, and health facility inventories. Consistent with the WHO framework for integrated malaria services delivery for pregnant women, integration was operationalized by linking these three levels to capture whether pregnant women received coordinated malaria-related services during ANC.

**Integrated malaria service delivery during ANC.** The concurrent presence of three malaria service components during an ANC encounter: (1) malaria prevention counseling on ITN use reported by the client, (2) provider capacity to manage malaria cases according to national guidelines, and (3) facility readiness for malaria diagnosis and treatment. These components reflect process and structural indicators of integrated service delivery derived from the client interview, provider interview, and facility inventory modules of the SPA survey.

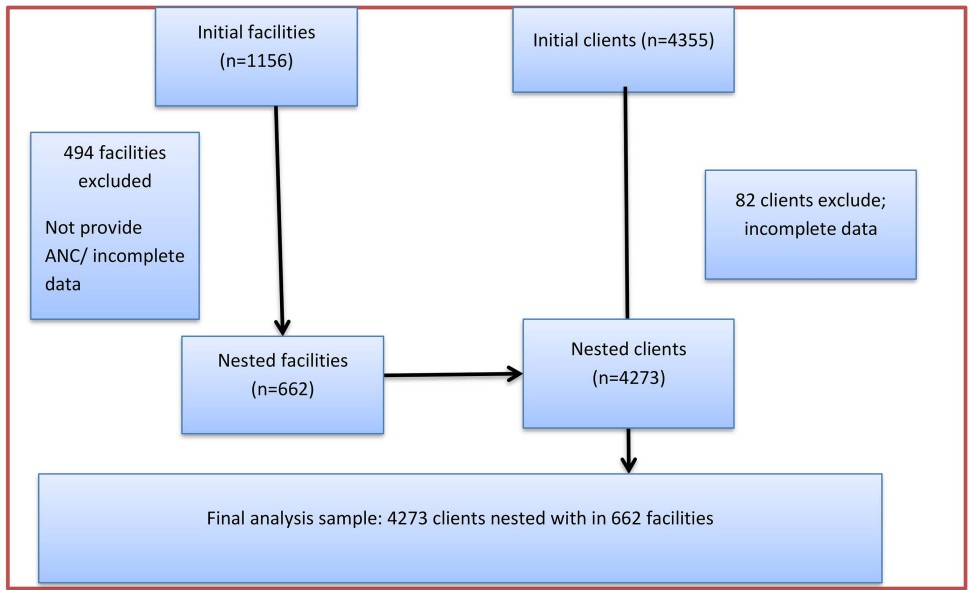

**Fig 1. Flow diagram of the sampling structure and dataset composition for the analysis of integrated malaria service delivery among pregnant women attending ANC in Ethiopia, 2021/22.**

**Indicator 1: Malaria prevention counseling (client-level).**  Malaria prevention counseling was measured using the client exit interview dataset. Pregnant women attending ANC were asked whether they received counseling on the use of insecticide-treated nets (ITNs) for malaria prevention. This variable was coded as:

- **Yes (1):** Client reported receiving counseling on ITN use

- **No (0):** Client did not receive counseling on ITN use

ITN counseling was selected as the preventive component because it is the standard malaria prevention intervention delivered within ANC services in Ethiopia, whereas ITN receipt typically occurs through community or household-level distribution campaigns. Intermittent preventive treatment in pregnancy (IPTp) is not part of Ethiopia's national malaria policy and was therefore not included.

**Indicator 2: Provider capacity for malaria case management (provider-level).**  Provider capacity to deliver malaria case management was assessed using the health care provider interview dataset. Providers were classified as having malaria case management capacity if they reported routinely providing malaria diagnosis and treatment services as part of their clinical responsibilities.

Providers meeting these criteria were coded as:

- **Adequate capacity (1)**

- **Inadequate capacity (0)**

**Indicator 3: Facility readiness for malaria diagnosis and treatment (facility-level).**  Facility readiness was assessed using the facility inventory dataset. Facilities were classified as malaria-ready if they had:

- At least one functional malaria diagnostic method available on the day of assessment (rapid diagnostic test and/or microscopy), and

- At least one first-line antimalarial medication in stock consistent with national treatment guidelines.

Facilities meeting both criteria were coded as:

- **Ready (1)**

- **Not ready (0)**

This indicator captures the structural capacity required to support integrated malaria service delivery during ANC.

**Construction of the integrated malaria service delivery variable.**  The three indicators were combined to generate a composite measure of integrated malaria service delivery, reflecting whether malaria prevention, diagnosis, and treatment services were delivered in a coordinated manner during ANC. Integration status was classified into four mutually exclusive categories:

- **Fully integrated:** All three components present

- **Partially integrated:** Any two of the three components present

- **Limited integrated:** Only one component present

- **Not integrated:** None of the components present

For regression analyses, the outcome variable was dichotomized as follows:
Integrated service delivery (1): all three service components were present.
No integrated service delivery (0): at least one of the three service components was not present.
This approach aligns with WHO guidance on integrated service delivery and reflects meaningful thresholds for coordinated malaria care during pregnancy.

**Study covariates.**  Covariates were selected *a priori* based on the WHO malaria in pregnancy service integration framework, existing literature, and data availability in the 2021/22 Ethiopia Service Provision Assessment. Variables were grouped into client-level and facility-level factors to reflect the hierarchical structure of the data.

**Client-level covariates.**  Client-level variables were extracted from the ANC client exit interview dataset and included:

- Maternal age group (15–24, 25–34, ≥ 35 years)

- Educational status (no formal education, primary(grades (1–8), secondary (9–12) or higher (diploma and above)

- Marital status (married vs. not married)

- Pregnancy status (gravida) (primigravida vs. multigravida)

- Number of ANC visits (1, 2, ≥ 3 visits)

- Receipt of insecticide-treated net (ITN) during ANC (yes/no)

- Knowledge of fever as a danger sign in pregnancy (yes/no)

- Payment for ANC services (yes/no)

**Facility-level covariates.**  Facility-level variables were obtained from the SPA facility inventory and provider interview datasets and included:

- Facility location (urban/rural)

- Facility ownership (public vs. private/NGO)

- Facility category (hospital vs. health center)

- Availability of malaria diagnostics (microscopy or rapid diagnostic test)

- Availability of first-line antimalarial drugs in line with national guidelines

- Facility malaria training: defined as the presence of at least one ANC service provider at the facility who reported receiving formal training on malaria diagnosis and treatment within the past 24 months (yes/no)

- Supervision received: defined as whether the facility or ANC provider reported receiving any external supervisory visit related to maternal or malaria services within the last six months (yes/no)

All facility- and provider-level variables were linked to clients through the facility identifier and treated as higher-level covariates in the multilevel analysis

**Operational definition.  Integrated malaria Service Delivery:**

Delivery of coordinated malaria- services during ANC, defined by the concurrent presence of malaria prevention counseling, provider capacity for malaria case management, and facility readiness for malaria diagnosis and treatment.

**Malaria burden regions** were operationally defined as a categorical variable with three levels: "High," "Moderate," and "Low." This classification was based on the endemicity and malaria transmission intensity observed across Ethiopia's administrative regions, aligned with the national malaria stratification [35].

- **High Burden Region:** Regions characterized by high prevalence/incidence and higher caseloads. These include Gambela, Benishangul-Gumuz, Amhara, Oromia, and Southern Nations, Nationalities, and Peoples' Region (SNNPR) regional states.

- **Moderate Burden Region:** Regions with moderate case numbers. These include Somalia, Sidama, and Afar regional states.

- **Low Burden Region:** Regions with sporadic or minimal malaria transmission. These include Addis Ababa, Dire Dawa city administrations, and Harari regional state.

## Data management and analysis

Missing data management was employed using different techniques. Variables with minimal missingness (<1) (e.g., diagnostic availability, supervision) were conservatively recoded, while those with moderate missingness (> 1) (e.g., guideline availability, training) were addressed using Multiple Imputation by Chained Equations (MICE) with logistic regression models [36]. The outcome variable was recomputed in each imputed dataset prior to modeling.

Health care provider data were first linked to the respective facilities and then aggregated to represent facility-level characteristics. The presence of multicollinearity among independent variables was assessed using Generalized Variance Inflation Factors (GVIF). All GVIF values were well below the common threshold of 5, indicating no significant multicollinearity issues.

Descriptive statistics were used to summarize the characteristics of the study population, with categorical variables presented as frequencies and percentages. A forest plot was used to display the regional distribution of malaria service integration among antenatal care (ANC) users. In addition, bar charts were used to illustrate the status of malaria service integration across regions.

Multilevel logistic regression analysis was employed to identify factors associated with the receipt of integrated malaria services delivery, accounting for the hierarchical structure of the data where clients were nested within facility. The analysis was conducted using statistical software R (version 4.3) with the *lme4* package for mixed-effects models.

We used the incremental value of adding different levels of predictors, three multilevel logistic regression models were compared: a null model (intercept only), a client-level model (model I), and a full model (combining client and facility-level predictors). The comparison of the multilevel logistic regression models demonstrated that the facility-level clustering accounted for a substantial proportion of the total variation in integrated malaria services delivery, as reflected by the high intra class correlation coefficient (ICC) of 0.934 in the null model. When client-level and facility-level predictors were sequentially added, the conditional $R^2$ remained high (0.907 in the final model), indicating that the random effects at the facility level continued to explain most of the variability. However, the marginal $R^2$ increased from 0.000 in the null model to 0.022 in the full model, showing that the inclusion of fixed predictors modestly improved the model's ability to explain variation in integration status. The root mean square error (RMSE) decreased slightly from 0.191 to 0.190, suggesting improved predictive performance. Model selection criteria further supported the final model; it had the lowest AIC indicating a superior balance of goodness of fit.

Adjusted Odds Ratios (AORs) with their corresponding 95% Confidence Intervals (CIs) and p-values were reported for fixed effects in the final model. We conducted bivariate analyses, and variables with a p-value ≤ 0.2 were retained for final modeling. A p-value of less than 0.05 was considered statistically significant.

## Ethical considerations

The 2022 Ethiopia SPA Survey received ethical clearance from the EPHI and the Institutional Review Board of ICF International. For this secondary analysis, anonyms and publicly available data were used, and no additional ethical approval and consent was required [37].

## Results

### Socio-demographic and client-level characteristics

The study analyzed data from 4,273 pregnant women who received ANC services in Ethiopia, based on the 2021/2022 Ethiopia SPA survey. The analysis of client-level characteristics provides insights into the socio-demographic profile and service utilization patterns of the participants.

The mean age of the client was 25.69 years (SD = 4.99), with the largest proportion—1,618 women (37.86%)—falling within the 25–29 years age group. Regarding educational status, 1,475 participants (34.52%) had attained secondary education. The majority of respondents were from urban areas 2,891 (67.65%), and most were multigravidas, accounting for 2,971 women (69.53%).

In terms of ANC utilization, nearly half 2,033 (47.58%) of the women had attended only one ANC visit. Alarmingly, a significant majority, 3,847 women (90%) were unaware that fever is a danger sign during pregnancy. Additionally, most participants 3,415 (79.89%) got ANC service free of payment.

Concerning malaria prevention, the findings reveal substantial service gaps. Only 123 women (2.88%) reported receiving insecticide-treated nets (ITNs) at the health facility, and just only 859 women (20%) received counseling on ITN use, a key intervention for malaria prevention during pregnancy (Table 1).

## The distribution of facilities and clients in Ethiopia

The distribution of clients were varied by regions, For example, the Afar region had the smallest sample size with 85 clients, while the Oromia region represented the largest portion of the study population with 1,176 clients (Table 2).

Facility-level characteristics provide critical insights into the healthcare system environment that pregnant women attended ANC visits where integrated malaria services were delivered. The vast majority of clients 4,157 (97.29%) received services from facilities that offered malaria case management services, including diagnosis and treatment. Additionally, 2,980 (69.74%) of the women received care at facilities equipped with malaria diagnostics (microscope or RDT). Hospitals accounted for the largest share of service locations 3,009 (70.42%). In terms of ownership, governmental

Table 1. Socio-demographic and client-level characteristics of pregnant women attending ANC in Ethiopia, 2021/22 National Service Provision Assessment (N = 4,273).

| Characteristic | Category | Frequency (N) | Percentage (%) |
|---|---|---|---|
| Maternal Age Group | <25 years | 1,618 | 37.86 |
| | 25–29 years | 1,627 | 38.08 |
| | >30 years | 1,028 | 24.06 |
| Education Level | No education | 911 | 21.32 |
| | Primary (grade 1–8) | 1,361 | 31.85 |
| | Secondary (grade 9–12) | 1,475 | 34.52 |
| | Higher (college and above) | 526 | 12.31 |
| Gravidity | 1 | 1,302 | 30.47 |
| | 2 | 1194 | 27.94 |
| | 3 or more | 1777 | 41.59 |
| ANC Visits | 1 | 2,033 | 47.58 |
| | 2 | 1,001 | 23.43 |
| | 3 | 630 | 14.74 |
| | 4 or more | 609 | 14.25 |
| Paid for ANC | No | 3,415 | 79.92 |
| | Yes | 858 | 20.08 |
| Knows Fever as Danger Sign | No | 3,847 | 90.00 |
| | Yes | 426 | 10.00 |
| Received ITN | No | 4,150 | 97.12 |
| | Yes | 123 | 2.88 |
| Counseled ITN use | No | 3,414 | 79.90 |
| | Yes | 859 | 20.10 |

**Table 2. Distribution of health facilities and ANC clients across regions in Ethiopia, 2021/22 National Service Provision Assessment Survey.**

| Region | Facilities Included | Total Clients |
|---|---|---|
| Afar | 24 | 85 |
| Amhara | 108 | 683 |
| Oromiya | 171 | 1376 |
| Somali | 41 | 220 |
| Benishangul | 21 | 115 |
| SNNP | 106 | 781 |
| Gambella | 35 | 115 |
| Harari | 17 | 80 |
| Addis Ababa | 52 | 379 |
| Dire Dawa | 27 | 126 |
| Sidama | 60 | 313 |
| **Total** | **662** | **4273** |

facilities dominated the service landscape, with 3,774 (88.32%) of women attending these institutions, while private or NGO facilities served a smaller portion (499; 11.68%).

Geographically, majority services were delivered in urban areas, where 2,891 (67.65%) of clients were seen. Notably, 3,070 women (71.85%) were seen at facilities located in high malaria burden regions (Table 3).

### Classification and distribution of malaria service delivery

As shown in Fig 2, the majority of pregnant women (53.3%) received ANC services in facilities classified as few integrated, where only one malaria service component was available.

An additional 37.3% of ANC clients accessed partially integrated. Only 7.9% of pregnant women attended ANC visits where integrated malaria services were delivered.Conversely, 1.5% of clients received ANC services in facilities with no malaria service integration (Fig 2).

### Integrated malaria ServiceDelivery and disparities across regions in Ethiopia

Nationally, only 7.9% (95% CI: 7.1%–8.7%) of pregnant women received ANC care visits where fully integrated malaria services were delivered.

Considerable regional disparities were observed in the delivery of integrated malaria services during ANC across Ethiopia. As presented in Fig 3, the proportion of clients receiving integrated malaria services exceeded the national average in several regions. The highest coverage was recorded in Benishangul Gumuz at 40.0% (95% CI: 31.5%–49.1%), followed by Dire Dawa at 23.8% (95% CI: 17.2%–32.0%), Gambella **at** 22.6% (95% CI: 15.9%–31.1%), and Harari **at** 18.8% (95% CI: 11.7%–28.7%).

Conversely, pregnant women who visited facilities in four regions exhibit integrated malaria service Delivery proportions considerably lower than the national average. Client visit health facilities in Addis Ababa registered the lowest integration proportions at 1.8% [95% CI: 0.9%, 3.8%], followed by Sidama at 3.5% [95% CI: 2.0%, 6.2%], Oromia at 4.7% [95% CI: 3.7%, 5.9%], and S.N.N.P. at 4.9% [95% CI: 3.6%, 6.6%]. The disparities highlight the statistical significance of these regional differences compared to the national average malaria integration (Fig 3).

### Multilevel logistic regression analysis

The multilevel logistic regression analysis model examined factors associated with the delivery of integrated malaria services among ANC clients, accounting for both client-level and facility-level influences. In model, where client- and

**Table 3. Facility-level characteristics related to integrated malaria service delivery during ANC among pregnant women in Ethiopia, 2021/22 National Service Provision Assessment Survey (N = 4,273).**

| Characteristic | Category | Frequency (N) | Percentage (%) |
|---|---|---|---|
| Malaria tx Services avai | No | 116 | 2.71 |
| | Yes | 4,157 | 97.29 |
| Malaria Test Lab avai | No | 1,293 | 30.26 |
| | Yes | 2,980 | 69.74 |
| Facility Category | Hospital | 3,009 | 70.42 |
| | Other Health Facility | 1,264 | 29.58 |
| Ownership | Governmental | 3,774 | 88.32 |
| | Private/NGO | 499 | 11.68 |
| Location | Urban | 2,891 | 67.65 |
| | Rural | 1,382 | 32.34 |
| Facility in Malaria Burden Region | Low | 585 | 13.68 |
| | Moderate | 618 | 14.47 |
| | High | 3,070 | 71.85 |

Facility malaria tx services" refers to the availability of malaria treatment services, while "malaria test lab availability" refers to the presence of laboratory capacity for malaria testing.

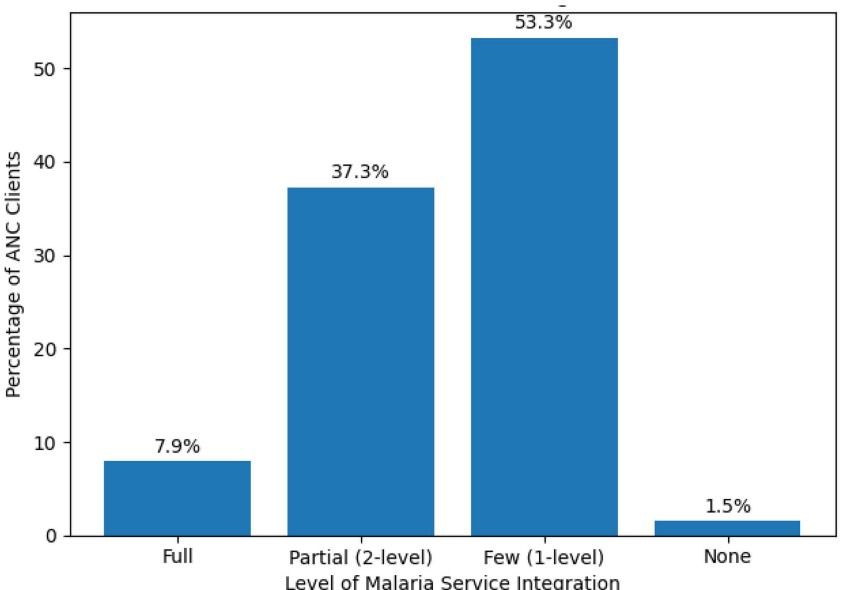

**Fig 2. Classification and distribution of integrated malaria service delivery during antenatal care among pregnant women in Ethiopia, 2021/22 National Service Provision Assessment Survey.**

facility-level factors were simultaneously included, five factors were found to be significantly associated with the integration of malaria services (p < 0.05).

At the client level, pregnancy status (gravida), number of ANC visits, and receipt of ITNs were significant predictors. Women with two previous pregnancies were 1.67 times more likely than primigravida women to attend visits in which integrated malaria services were delivered (AOR = 1.67, 95% CI: 1.06–2.62).

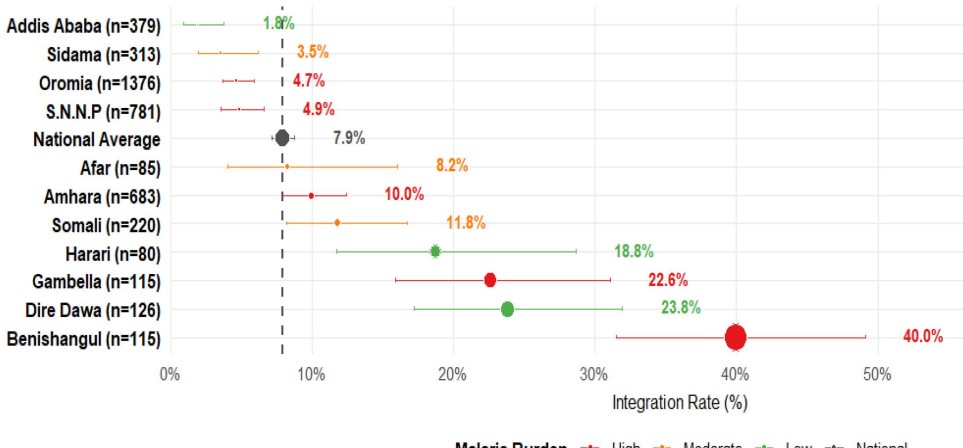

**Fig 3. Forest plot showing the proportion of integrated malaria service delivery during ANC by region in Ethiopia, 2021/22 National Service Provision Assessment Survey.**

Similarly, attendance at three or more ANC visits was associated with higher odds of receiving care during visits in which fully integrated malaria services were delivered, compared with attendance at only one ANC visit (AOR = 1.58; 95% CI: 1.04–2.40). Receipt of an ITN during ANC was also positively associated with integrated malaria service delivery (AOR = 2.81; 95% CI: 1.29–6.12).

At the facility level, providers training on malaria and facility location were significantly associated with service integration delivery.

Clients attending facilities with malaria-trained providers had significantly higher odds of receiving integrated malaria services during ANC than those attending facilities without trained providers (AOR = 4.24; 95% CI: 1.80–10.00). Likewise, clients receiving ANC in rural facilities were more likely to experience integrated malaria service delivery than those in urban facilities (AOR = 2.73; 95% CI: 1.04–7.19). In contrast, knowledge of fever as a danger sign, facility supervision, facility category, and ownership were not significantly associated with integrated malaria service delivery.

(Table 4).

## Discussion

This study utilized a nationally representative dataset from the 2021/22 Ethiopia SPA survey to assess the level of integration of malaria services delivery during ANC and to identify the multifaceted client and facility-level factors influencing this delivery.

The delivery of integrated malaria services into ANC remains critically limited in Ethiopia. This low figure underscores missed opportunities for delivering comprehensive malaria interventions at a critical point of maternal care. This indicates a substantial gap in the country's efforts to safeguard pregnant women from malaria and achieve its ambitious 2030 malaria elimination goals, aligning with the observed re-emergence of malaria as a top health concern despite previous progress [38].

The most common pattern observed was limited integration, accounting for 53.3% of all ANC visits. In these cases, although facilities were capable of providing malaria diagnosis and treatment, there was a lack of provider-level engagement or client counseling on malaria prevention. This finding indicates that infrastructure alone is insufficient to guarantee service delivery, and highlights the need to strengthen the human resource and communication components of ANC visits. The findings is consistent with the analysis of service integration delivery in SSA [39].

**Table 4. Multilevel determinants of integrated malaria service delivery during ANC among pregnant women in Ethiopia, 2021/22 National Service Provision Assessment Survey.**

| Variables | Category | Model 1 AOR (95% CI) | Final Model AOR (95% CI) |
|---|---|---|---|
| Gravida | 1 (Ref) | 1 | 1 |
| | 2 | **1.61 (1.05–2.45)** | **1.67 (1.06–2.62)** |
| | 3 or more | 1.25 (0.83–1.90) | 1.33 (0.86–2.05) |
| ANC visits | 1 (Ref) | 1 | 1 |
| | 2 | 1.15 (0.78–1.71) | 1.30 (0.84–1.99) |
| | 3 or more | 1.29 (0.89–1.88) | **1.58 (1.04–2.40)** |
| Knows fever danger sign | No (Ref) | 1 | 1 |
| | Yes | 1.11 (0.70–1.76) | 1.16 (0.65–2.07) |
| Received ITN | No (Ref) | | |
| | Yes | **2.25 (1.22–4.15)** | **2.81 (1.29–6.12)** |
| Supervision received | No (Ref) | 1 | 1 |
| | Yes | — | 1.44 (0.82–2.50) |
| Facility training malaria | No (Ref) | 1 | 1 |
| | Yes | — | **4.24 (1.80–10.0)** |
| Facility category (Other) | Hospital | 1 | 1 |
| | Others | — | 1.99 (0.77–5.12) |
| Ownership of facility | Government | | |
| | Private/NGO | — | 1.81 (0.53–6.16) |
| Location of facility | Urban | 1 | 1 |
| | Rural | — | **2.73 (1.04–7.19)** |

These findings reflect systemic challenges in aligning policy, capacity, and practice. Despite WHO recommendations to integrate malaria services—particularly IPTp, case management, and promotion and use of ITN —within ANC [17], the Ethiopian context lacks full policy adoption, particularly the absence of IPTp-SP and ITN provision at ANC in national guidelines [24]. Furthermore, the study revealed significant regional disparities in delivery of integration services across Ethiopia.

The delivery of integrated malaria services during ANC visits was markedly higher in some regions, indicating significant regional variation relative to the national average. The heterogeneity observed in malaria service integration across regions may stem from various factors. For regions with historically low malaria burden, such as Addis Ababa, the low service integration might be explained by an assumed minimal risk, leading to reduced programmatic focus or awareness regarding malaria in pregnancy. However, the critically low integration delivery was observed in regions like Oromia and S.N.N.P are particularly concerning given that these are recognized as high malaria burden regions. This paradox suggests that despite a high epidemiological need, the health systems in these areas may be overwhelmed, prioritizing acute case management over the integration of preventive and diagnostic services into routine ANC. This could also indicate a lack of specific resource allocation or targeted strategies for MiP within these high-burden contexts. While the delivery of integrated services in regions like Amhara showed a slightly higher (10%) than the national average, it remains critically low when compared to WHO recommendations, which advocate for every pregnant woman to receive comprehensive integrated services during ANC [40].

This overall scenario strongly indicates that the Ethiopian health system might currently place greater emphasis on general population malaria prevention and treatment strategies, with a notable absence of specific, robust guidelines and programs designed to support the seamless integration of malaria services into ANC. This is further supported by

qualitative findings from Ethiopia, which highlight the absence of malaria-specific interventions tailored for pregnant women, leaving them vulnerable to inadequate protection and care [29].

The multilevel logistic regression analysis revealed important determinants of malaria service integration among pregnant women attending antenatal care (ANC).

Women with two previous pregnancies (Gravida 2) and those with three or more ANC visits were more likely to receive care during visits in which fully integrated malaria services were delivered. This could suggest that multiparous women, having navigated the healthcare system before, are more adept at accessing comprehensive care, or that providers are more likely to offer extensive services to women with established records. The findings is consistent with the previous study established, pregnant women previous service experiences are favorable contributor for service integration [41].

However, this finding contrasts with the widely accepted recommendation that primigravidae should be prioritized for malaria prevention due to their greater susceptibility to malaria infection and its complications. The immune-naïve status of first-time pregnant women places them at higher risk for severe disease and adverse pregnancy outcomes [42,43].

On the other side, those attending three or more ANC visits were 1.48 times more likely to receive care during visits in which fully integrated malaria services were delivered. It's plausible that as women engage more consistently with ANC services, opportunities for integrated malaria services naturally increase due to repeated interactions with the health system. These findings align with prior finding indicating that increased contact with health facilities enhances access to health education, malaria screening, and preventive interventions [10,44]. The association may also reflect the cumulative benefit of repeat ANC exposure in promoting integrated care.

The receiving an ITN during ANC were 2.81 times more likely receiving care during visits in which fully integrated malaria services were delivered. This finding suggests the provision of an ITN may serve as an entry point of a more comprehensive, integrated approach to malaria prevention and control within ANC. It is possible that facilities prioritizing ITN distribution are also more likely to be implementing other components of integrated malaria services, such as counseling and diagnosis. This finding is in line with previous evidence showing that integrated antenatal and malaria services are positively associated with improved ITN access [45].

At the facility level, this study uncovered important disparities in the delivery of integrated malaria services for pregnant women across geographic and epidemiologic contexts.

The finding reveled that clients who visit facilities where providing malaria training were significantly more likely to received care during visits in which fully integrated malaria services were delivered. This aligns with global recommendations emphasizing the need for skilled healthcare workers to deliver quality and integrated care [27,39].

Pregnant women who got service in rural facilities were significantly more likely to received care during visits in which fully integrated malaria services were delivered compared to their urban counterparts. This finding may reflect the targeted public health strategies implemented in rural areas, where maternal and malaria-related mortality have historically been higher [46]. Additionally, rural health facilities may experience lower patient volumes, allowing for longer consultation times and more opportunities for integrated service delivery [47]. However, there are contradict findings showed the rural health facilities are limited in accessing of qualified professional, essential commodities that affect the delivery of integrated service [48].

### Strengths and limitations

This study's strengths include its use of nationally representative data from the recent 2021/22 SPA survey, providing robust generalizability. The large sample size and the application of multilevel logistic regression adequately addressed the hierarchical nature of the data, minimizing bias from clustering effects. Furthermore, the comprehensive inclusion of client and facility-level variables allowed for a nuanced understanding of the complex determinants.

Nevertheless, certain limitations should be acknowledged. Its cross-sectional design precludes the establishment of causal relationships between the identified factors and service integration and the SPA survey may not fully align with

Ethiopia seasonal variation in malaria transmission; this could have influence the malaria service integration practice. In addition, the reliance on facility-based surveys and client exit interviews may be subject to recall bias or social desirability bias. Moreover, the composite operational definition of "integrated malaria services delivery" was stringent, which might contribute to the low overall integration rate. Finally, despite the use of multilevel modeling to account for clustering of clients within facilities, residual unmeasured facility- or community-level factors may still influence the observed associations.

Further research using regionally powered surveys or routine health information system data is recommended to generate more robust subnational estimates and to better understand geographic inequities in malaria service integration with ANC services.

## Conclusion

The delivery of integrated malaria services into antenatal care in Ethiopia remains critically low, marked by significant regional disparities and multilevel determinants. Key factors influencing service integration delivery include gravida, ANC visit frequency, and the provision of ITNs at the client level. At the facility level, malaria-specific training and rural location significantly impacted service integration. Addressing these complex challenges necessitates a multi-pronged approach: (i) Policy and Service Readiness: National and regional health authorities should prioritize ensuring that all ANC facilities are equipped with malaria diagnostics, treatment, and trained providers to delivered consistent and integrated services; (ii) Capacity building: continuous in-service training on MiP, including counseling on ITN use and case management, should be integrated into routine professional development programs; (iii) Equity Service delivery: regional disparities highlights the needs of intensified support in high-burden and underserved areas through targeted resource allocation, supportive supervision, and capacity building; (iv) Client Engagement: strengthening demand side intervention through promoting early and consistent ANC attendance and revising national polices to formally integrate ITN distribution into ANC follow up; (v) Health System adaptation: successful rural facility practices should be adapted and scaled to urban and high-burden regions where integration lags and (vi) Future research: Further studies should explore underlying causal pathways to fully understand the determinants of delivery of malaria service integration for pregnant women in Ethiopia.

## Acknowledgments

The authors thank the DHS Program for granting access to the 2022 Ethiopia SPA dataset. We also acknowledge the Ethiopian Public Health Institute and the Ministry of Health for facilitating the survey. The authors also acknowledged Bahir Dar University for the financial and technical supports.

## Author contributions

**Conceptualization:** Kassawmar Angaw Bogale, Kindie Fentahun Muchie.

**Formal analysis:** Kassawmar Angaw Bogale.

**Methodology:** Kassawmar Angaw Bogale, Kindie Fentahun Muchie, Mulusew Andualem Asemahagn, Muluken Azage Yenesew.

**Software:** Kindie Fentahun Muchie, Muluken Azage Yenesew.

**Supervision:** Kassahun Alemu, Mulusew Andualem Asemahagn, Hailemariam Awoke Engedaw.

**Validation:** Kassahun Alemu, Kindie Fentahun Muchie.

**Visualization:** Hailemariam Awoke Engedaw.

**Writing – original draft:** Kassawmar Angaw Bogale.

**Writing – review & editing:** Kassahun Alemu, Kindie Fentahun Muchie, Mulusew Andualem Asemahagn, Hailemariam Awoke Engedaw, Muluken Azage Yenesew.

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
