## [Decision Letter · Decision Letter 0]

19 Aug 2025

Dear Dr. Bogale,

plosone@plos.org. . . . A rebuttal letter that responds to each point raised by the academic editor and reviewer(s). You should upload this letter as a separate file labeled 'Response to Reviewers'.A marked-up copy of your manuscript that highlights changes made to the original version. You should upload this as a separate file labeled 'Revised Manuscript with Track Changes'.An unmarked version of your revised paper without tracked changes. You should upload this as a separate file labeled 'Manuscript'.

We look forward to receiving your revised manuscript.

Kind regards,

Khin Thet Wai, MBBS, MPH, MA

Academic Editor

PLOS ONE

Journal Requirements:

Additional Editor Comments :

Please do major revisions as required.

Reviewers' comments:

Reviewer's Responses to Questions

**Comments to the Author**

1. Is the manuscript technically sound, and do the data support the conclusions?

Reviewer #1: Partly

Reviewer #2: Partly

2. Has the statistical analysis been performed appropriately and rigorously?

Reviewer #1: Yes

Reviewer #2: Yes

3. Have the authors made all data underlying the findings in their manuscript fully available?

Reviewer #1: Yes

Reviewer #2: Yes

4. Is the manuscript presented in an intelligible fashion and written in standard English?

Reviewer #1: Yes

Reviewer #2: Yes

Reviewer #1: This manuscript investigates the uptake of integrated malaria services and its determinants among pregnant women in Ethiopia. Given the substantial burden of malaria during pregnancy, this is a highly significant topic. To enhance the quality of this manuscript, the following suggestions are put forward:

1. In the "Study Population and Sampling" section, the authors should elaborate on the reasons for excluding 494 facilities within the text.

2. The "Variables and Measurements" section could benefit from reorganization to eliminate redundancy. For example, it is recommended that the paragraph describing the outcome variable be merged with its operational definition.

3. In this study, service integration is defined based on three criteria: counseling on insecticide-treated nets (ITNs), the provider's ability to deliver malaria case management in accordance with national guidelines, and the availability of malaria diagnosis and treatment services at the facility. The authors need to provide more specific details on how these variables were measured using the SPA. Additionally, it would be valuable to explain why the focus was placed on counseling on ITNs rather than the receipt of ITNs or other preventive measures such as IPTp. Furthermore, is there a theoretical foundation for constructing such a combined measurement of service integration?

4. The authors report on malaria service integration and disparities across regions in Ethiopia. It is important to clarify whether the SPA data can generate representative results for each individual region.

5. Only a small number of variables mentioned earlier are included in Table 4. Did the authors exclude variables that did not yield significant estimates in the model? Moreover, the difference between Model 1 and the final model should be clearly explained. The authors are advised to report all variables included in both models.

Reviewer #2: Thank you for the opportunity to review this interesting article that examines facility- and client-level factors associated with receipt of integrated MiP services during ANC in Ethiopia. Below are major and minor recommendations to help strengthen this important manuscript.

Major recommendations:

Methods: Consistency in use of the language for the model used, i.e. multilevel mixed effects multivariable logistic regression, is needed. What are the fixed versus random effects in the model?

Did you perform a complete case analysis on your analytic sample? How did you handle missing data?

How are health care provider (interviews?) handled in your analyses? Did you aggregate them in some way at the facility level? This is confusing throughout the paper and makes interpretability challenging.

Is there a reason for listing the Outcome Variable and then the Operational Definition separately? These could be combined as the same thing. It would be helpful to know where each of these variables comes from in the SPA questionnaires. Is there a specific question that corresponds to each? Or are they constructed as composites of questions?

Where do data for the malaria regions originate from? SPA or other?

What was your approach to variable selection and inclusion in the adjusted models, beyond model building for multilevel models (i.e. did you select variables a priori based on the literature, based on some cutpoint, etc? (Figure 4)

Why was region not included in the models?

Results/Figures: All figures need some updating to make them interpretable. A map, perhaps of malaria endemicity zones (High, Medium, Low) and regions would be helpful; better yet if it shows the distribution of facilities.

Fig 1 would benefit from a redesign to show how clients cluster within facilities, rather than showing facilities and clients side by side.

Fig 2 would benefit from reorganizing the key from full to none. The labels as presently listed along the y-axis make this figure difficult to interpret, perhaps due to lack of specificity. Consider using the same language in prose as in the figure to help clarify. Please also clarify whether you are presenting data at the client-level or the facility-level.

Fig 3 Is the National Average (7.9%) an actual average? I understand it to be the percent of clients who received care at a facility that met the 3-pronged criteria for integrated MiP in ANC, for clients included in your analytic sample. Also, consider relabeling these results - they aren't really a rate. Perhaps frequency would be a better fit.

The section on model assessment should either be reduced to a line or two under the results of the model, or put in supplemental files.

Discussion:

Consider further exploring in greater detail the effect of cross-sectional data on your results. Were SPA questionnaires administered all at once, or on a rolling basis region-by-region? How does that align / not align with malaria seasonality in Ethiopia? How does timing of client interviews potentially effect integration of MiP in ANC

You mention the health extension worker program that includes an element of bed nets and counseling. Please consider going into greater detail about how this could affect your results. If ITNs are not required per national policy as part of MiP services - is this something that is delegated to the CHEWs? Why did you utilize ITN measures but not IPTp, as neither are part of national policy?

What was the presence of international aid in Ethiopia, by region, at the time of survey? Could that have had a demonstrable impact on your results?

Consider adding a section (that is supported by your findings) that makes public health recommendations/ addresses implications for national policy and/or programming

Minor recommendations:

1) needs a thorough copy-edit for typos and grammar

2) double check font type and size in tables and figures (e.g. table 1)

3) acronyms / abbreviations only need to be spelled out at first use rather than repeatedly throughout the manuscript

.

Reviewer #1: **Yes:** Di LiangDi LiangDi LiangDi Liang

Reviewer #2: **Yes:** Elizabeth H LeeElizabeth H LeeElizabeth H LeeElizabeth H Lee

---

## [Author Response · Author response to Decision Letter 1]

5 Sep 2025

Dear Khin Thet Wai, we revised the manuscript according to the reviewers' suggestions and the editor's recommendation.

---

## [Decision Letter · Decision Letter 1]

16 Jan 2026

Dear Dr. Bogale,

plosone@plos.org. . . . A letter that responds to each point raised by the academic editor and reviewer(s). You should upload this letter as a separate file labeled 'Response to Reviewers'.A marked-up copy of your manuscript that highlights changes made to the original version. You should upload this as a separate file labeled 'Revised Manuscript with Track Changes'.An unmarked version of your revised paper without tracked changes. You should upload this as a separate file labeled 'Manuscript'.

We look forward to receiving your revised manuscript.

Kind regards,

Khin Thet Wai, MBBS, MPH, MA

Academic Editor

PLOS One

Journal Requirements:

Additional Editor Comments:

There are flaws in methods and results sections especially for multiple statistical models. Please do major revisions as required.

Reviewer's Responses to Questions

**Comments to the Author**

Reviewer #1: (No Response)

Reviewer #2: (No Response)

Reviewer #3: (No Response)

2. Is the manuscript technically sound, and do the data support the conclusions?

Reviewer #1: Yes

Reviewer #2: Yes

Reviewer #3: (No Response)

3. Has the statistical analysis been performed appropriately and rigorously?

Reviewer #1: Yes

Reviewer #2: Yes

Reviewer #3: (No Response)

4. Have the authors made all data underlying the findings in their manuscript fully available?

Reviewer #1: Yes

Reviewer #2: Yes

Reviewer #3: Yes

5. Is the manuscript presented in an intelligible fashion and written in standard English?

Reviewer #1: Yes

Reviewer #2: Yes

Reviewer #3: No

Reviewer #1: Thanks the authors for addressing my previous comments. I still have one concern for this manuscript. Although SPA is a nationally representative dataset, it is not safe to assume that the national-level findings also apply to people at the local level. SPA may not be able to generate regionally representative estimates of service readiness and integration. Thus, the authors should be careful about the interpretation of regional-level findings in Figure 3 and the findings on page 13-14.

Reviewer #2: Thank you for your careful attention to initial comments. It appears that some of the figures (e.g. Fig 2) may still require attention based on the prior feedback. Additionally, I would recommend providing more information about study covariates in the methods section. Additional clarity about use of the outcome variables (integration y/n and the stratified integration variable) is needed. There are two places where the number of levels / models appear to be incorrectly noted. pg 9: Model performance was compared across the four models using various indices (line 222 in revisions); and pg 7: Variables were categorized into three hierarchical levels (line 161 in revisions).

The authors may also wish to present results stratified by malaria burden area classification or urban/rural status, as a major question remains whether these variables may modify likelihood of receipt of integrated services.

Reviewer #3: Thank you to the editor and authors for giving me the opportunity to read this manuscript. I have noted that the authors revised the paper based on other reviewers’ comments. However, I believe the manuscript still requires major revisions, particularly regarding the logical flow, operational definitions of outcome variables, and data analysis. My main comments are as follows:

1. The title suggests an assessment of service uptake, yet there is no mention or analysis of uptake throughout the manuscript.

2. Line 31 mentions “unborn,” whereas line 59 refers to “newborns.” Please ensure consistent terminology.

3. There are several typographical and grammatical errors throughout the manuscript (e.g., lines 50–52).

4. Terms such as “integrated malaria services” and “malaria service integration” are used inconsistently. Please clarify what these mean. Are these based on the authors’ definitions or existing frameworks? Has such integration already been implemented in Ethiopia? If so, when and how? Otherwise, low uptake or limited-service provision may be considered expected rather than exceptional.

5. Line 109: It is unclear whether this is a secondary data analysis or a cross-sectional study.

6. Figure 1 requires improvement for clarity and logical flow. For example, the arrows from nested clients to facilities appear misplaced.

7. Lines 154–155: The text refers to “danger sign knowledge,” but the data reflect only fever.

8. Lines 164–168: Please explain how the three indicators were derived. Were they from the original survey, or defined by the authors? If self-reported by participants, they may be biased and not reflect provider implementation. For indicators 2 and 3, specify the criteria used for classification.

9. In Table 1, font styles are inconsistent. Clarify the meaning of “primary,” “secondary,” and “higher” education. The proportion of participants who “received ITNs” is lower than those “counselled on ITN use.”

10. Table 2 and its corresponding text are not informative. They appear to describe sampling methods rather than findings. Did the original study use unequal sampling across facilities?

11. In Table 3, please define “facility malaria services” and “malaria test lab avai.”

12. Lines 265–270: Present the statistical results for the three main indicators. Lines 276–278 are unclear and need clarification.

13. Table 4 should be expanded. The title is vague. Include cross-tabulated data and corresponding simple logistics regression results (COR etc.) to substantiate the findings. If multiple models were tested, include all results as supplementary material. Variables such as “supervision received” and “facility malaria training,” which appear here but not in prior tables, must be defined and linked to earlier sections.

14. I have not reviewed the Discussion in detail, but the current version is lengthy and repetitive, focusing excessively on “integration of malaria services.” This section should be shortened and made more concise, while the part discussing associated factors should be expanded with references to more relevant studies.

.

Reviewer #1: **Yes:** Di LiangDi LiangDi LiangDi Liang

Reviewer #2: No

Reviewer #3: No

---

## [Author Response · Author response to Decision Letter 2]

21 Jan 2026

We have revised the manuscript based on the feedback.

---

## [Decision Letter · Decision Letter 2]

4 Feb 2026

Dear Dr. Bogale,

Thank you for submitting your manuscript to PLOS ONE. After careful consideration, we feel that it has merit but does not fully meet PLOS ONE’s publication criteria as it currently stands. Therefore, we invite you to submit a revised version of the manuscript that addresses the points raised during the review process.

We look forward to receiving your revised manuscript.

Kind regards,

Khin Thet Wai, MBBS, MPH, MA

Academic Editor

PLOS One

Journal Requirements:

Reviewer's Responses to Questions

**Comments to the Author**

Reviewer #1: All comments have been addressed

Reviewer #2: All comments have been addressed

Reviewer #3: (No Response)

2. Is the manuscript technically sound, and do the data support the conclusions?

Reviewer #1: Yes

Reviewer #2: Yes

Reviewer #3: No

3. Has the statistical analysis been performed appropriately and rigorously?

Reviewer #1: Yes

Reviewer #2: Yes

Reviewer #3: No

4. Have the authors made all data underlying the findings in their manuscript fully available?

Reviewer #1: Yes

Reviewer #2: Yes

Reviewer #3: Yes

5. Is the manuscript presented in an intelligible fashion and written in standard English?

Reviewer #1: Yes

Reviewer #2: No

Reviewer #3: No

Reviewer #1: (No Response)

Reviewer #2: Needs a typographical / grammar review by authors. There are many instances where additions / revisions now read with grammatical errors. For example, what is "few integration"?

Reviewer #3: Thank you to the authors for revising the manuscript. However, I believe the revisions remain minimal and inadequate in addressing many of the previous comments, partly due to the incorrect referencing of line numbers. I have listed several concerns below. I have not yet reviewed the Abstract, Background, Discussion, or Conclusions.

1. In the tracked-changes version (Lines 211–213), the authors state that the outcome variable was dichotomized as 1 and 0, where “0” indicates that at least one of the three service components was not present. Please clarify this statement. What do you mean by this definition, and how were other possible categories, such as “partially integrated” or “few integrated”, classified or handled?

2. Based on the operational definitions provided, the outcomes seem to evaluate facility performance rather than service uptake by participants. Specifically, in questions 2 and 3 on integrated services (Lines 258–260), the authors describe the presence of malaria prevention counseling, provider capacity for malaria case management, and facility readiness for malaria diagnosis and treatment. These indicators relate to service availability and provider capacity, not the uptake of malaria services by clients. Participants seeking ANC at facilities are already engaging in service uptake. The inability to receive malaria-related services likely reflects provider-side constraints rather than client behavior, meaning the study may not accurately measure “uptake.”

3. Line 296 mentions the use of VIF, while Line 333 refers to excluding variables with p < 0.2. How were these criteria applied when they potentially conflict with each other?

4. Many minor comments from the first review round remain unresolved, including: improvement of Figure 1, inconsistent font styles in Table 1, the limited relevance of Table 2 as a stand-alone table, unclear variable names in Table 3, insufficient expansion of Table 4 to include cross-tabulated data, and lack of improvement in the overall discussion flow.

.

Reviewer #1: No

Reviewer #2: No

Reviewer #3: No

---

## [Author Response · Author response to Decision Letter 3]

18 Feb 2026

Thank you for the opportunity to further revise our manuscript and for the continued constructive feedback from the reviewers.

We have carefully addressed all remaining concerns raised in this round of review. In particular:

• The outcome variable has been clarified conceptually and analytically to distinguish between descriptive integration categories and the dichotomized regression outcome.

• The interpretation of “integrated malaria service uptake” has been refined to reflect functional receipt of integrated services during ANC encounters rather than facility readiness alone.

• The model-building strategy has been clarified, including the sequential use of p-value screening and multicollinearity diagnostics (VIF).

• Figures and tables have been revised for clarity, consistency, and interpretability, including redesign of Figure 1 and expansion of Table 4.

• Language and terminology throughout the manuscript have undergone a thorough editorial review to correct grammatical inconsistencies and improve readability.

• The Discussion section has been streamlined to improve logical flow and better situate findings within existing literature.

We believe these revisions have substantially strengthened the methodological transparency, conceptual clarity, and overall presentation of the manuscript.

Thank you again for your guidance and consideration.

Sincerely,

Kassawmar Angaw Bogale

---

## [Decision Letter · Decision Letter 3]

7 Mar 2026

Dear Dr. Bogale,

plosone@plos.org. . . . A letter that responds to each point raised by the academic editor and reviewer(s). You should upload this letter as a separate file labeled 'Response to Reviewers'.A marked-up copy of your manuscript that highlights changes made to the original version. You should upload this as a separate file labeled 'Revised Manuscript with Track Changes'.An unmarked version of your revised paper without tracked changes. You should upload this as a separate file labeled 'Manuscript'.

We look forward to receiving your revised manuscript.

Kind regards,

Khin Thet Wai, MBBS, MPH, MA

Academic Editor

PLOS One

Journal Requirements:

Additional Editor Comments:

Please do revisions as required.

Reviewer's Responses to Questions

**Comments to the Author**

Reviewer #2: (No Response)

Reviewer #3: (No Response)

2. Is the manuscript technically sound, and do the data support the conclusions?

Reviewer #2: Yes

Reviewer #3: No

3. Has the statistical analysis been performed appropriately and rigorously?

Reviewer #2: Yes

Reviewer #3: No

4. Have the authors made all data underlying the findings in their manuscript fully available?

Reviewer #2: Yes

Reviewer #3: Yes

5. Is the manuscript presented in an intelligible fashion and written in standard English?

Reviewer #2: No

Reviewer #3: No

Reviewer #2: 1) It does not appear a thorough grammar review was conducted as was recommended previously. For example, (and this is one of many instances): "The finding is consistent with pervious study that the association of integrated antenatal and malaria service quality scores with ITN access [44]." I cannot tell what the authors are trying to state about reference 44 in relation to their findings based on the way this sentence currently reads.

2) I agree with reviewer two that the terminology 'service uptake' is challenging here because the authors are constructing their outcome based on structural and process factors. This seems much more in line with Service Availability and Readiness, not whether women necessarily used the services (e.g. for those who received an ITN, did they employ use of an ITN more often following receipt counseling/education). Indeed, the operational definition is: "Receipt of coordinated malaria-related services during ANC, defined by the concurrent presence of malaria prevention counseling, provider capacity for malaria case management, and facility readiness for malaria diagnosis and treatment. The authors should consider reframing their terminology to be a more accurate reflection of what is actually being measured. How are these constructs used and defined in the published literature?

Reviewer #3: Upon my quick reassessment of the revised manuscript, I find that the revisions are superficial and do not adequately address the concerns raised in the previous round of review. In many instances, no substantive changes have been made. Therefore, my comments remain the same as before.

.

Reviewer #2: No

Reviewer #3: No

---

## [Author Response · Author response to Decision Letter 4]

23 Mar 2026

Dear editor and reviewers,

Thank you very much for reviewing the manuscript and forward suggestions. We have revised the manuscript and addressed the comments.

Sincerely,

Kassawmar.

---

## [Editor Report · Decision Letter 4]

24 Mar 2026

Integrated Malaria Service Delivery and Its Determinants among Pregnant Women in Ethiopia: Multi level Analysis of 2021/22 National Service Provision Assessment Survey

PONE-D-25-38971R4

Dear Dr. Bogale,

We’re pleased to inform you that your manuscript has been judged scientifically suitable for publication and will be formally accepted for publication once it meets all outstanding technical requirements.

Kind regards,

Khin Thet Wai, MBBS, MPH, MA

Academic Editor

PLOS One
---

## [Editor Report · Acceptance letter]

PONE-D-25-38971R4

PLOS One

Dear Dr. Bogale,

I'm pleased to inform you that your manuscript has been deemed suitable for publication in PLOS One. Congratulations! Your manuscript is now being handed over to our production team.

Kind regards,

on behalf of

Dr. Khin Thet Wai

Academic Editor

PLOS One